# Nature versus Humans in Coastal Environmental Change: Assessing the Impacts of Hurricanes Zeta and Ida in the Context of Beach Nourishment Projects in the Mississippi River Delta

**Qiang Yao** [1] , **Marcelo Cancela Lisboa Cohen** [1,2,*] , **Kam-biu Liu** [1] , **Adriana Vivan de Souza** [2]
**and Erika Rodrigues** [1,2]

1   Department of Oceanography and Coastal Sciences, College of the Coast and Environment,
    Louisiana State University, Baton Rouge, LA 70803, USA; qyao4@lsu.edu (Q.Y.); kliu1@lsu.edu (K.-b.L.);
    erikarodrigues@ufpa.br (E.R.)
2   Graduate Program of Geology and Geochemistry, Federal University of Pará, Av. Perimentral 2651,
    Terra Firme, Belém 66077-530, PA, Brazil; adriana.vivan@terra.com.br
*   Correspondence: mcohen@ufpa.br

**Abstract:** Hurricanes are one of the most devastating earth surface processes. In 2020 and 2021, Hurricanes Zeta and Ida pounded the Mississippi River Delta in two consecutive years, devastated South Louisiana, and raised tremendous concerns for scientists and stakeholders around the world. This study presents a high-resolution spatial-temporal analysis incorporating planialtimetric data acquired via LIDAR, drone, and satellite to investigate the shoreline dynamics near Port Fourchon, Louisiana, the eye of Ida at landfall, before and after the beach nourishment project and recent hurricane landfalls. The remote sensing analysis shows that the volume of the ~2 km studied beachfront was reduced by 240,858 m³ after consecutive landfalls of Hurricanes Zeta and Ida in 2020 and 2021, while 82,915 m³ of overwash fans were transported to the backbarrier areas. Overall, the studied beach front lost almost 40% of its volume in 2019, while the average dune crest height was reduced by over 1 m and the shoreline retreated ~60 m after the two hurricane strikes. Our spatial-temporal dataset suggests that the Louisiana Coastal Protection and Restoration Authority's (CPRA's) beach nourishment effort successfully stabilized the beach barrier at Port Fourchon during the hurricane-quiescent years but was not adequate to protect the shoreline at the Mississippi River Delta from intense hurricane landfalls. Our study supports the conclusion that, in the absence of further human intervention, Bay Champagne will likely disappear completely into the Gulf of Mexico within the next 40 years.

**Keywords:** Hurricane Ida; Hurricane Zeta; LIDAR; Mississippi River Delta; satellite images; drone

## 1. Introduction

Hurricanes are one of the most devastating earth surface processes, causing over 10,000 deaths around the world each year [1]. For example, Hurricane Katrina (a Category 3 on the Saffir–Simpson Scale) made landfall near the Mississippi River Delta (MRD) on 29 August 2005 and killed over 1800 people in the USA [2]. In October 2020, another major hurricane, Zeta (Category 3), made landfall near the MRD. Ten months after Zeta, Hurricane Ida (Category 4) devastated the MRD again on the 16th anniversary of Hurricane Katrina. Along with Hurricane Laura (August 2020) and the Last Island Hurricane (1856), Ida was among the strongest hurricanes that struck the Louisiana coast in the last two centuries [3]. It also marks two major hurricane landfalls in Louisiana in two consecutive years (2020 and 2021) for the first time since ~1850s [4]. Moreover, instrumental records show that 90 hurricanes have made landfall near the Louisiana coast since the 1850s. Among them, 32 were major hurricanes (Categories 3–5) [4]. Thus, there is a vital and

urgent need to document the geomorphological impacts of such extreme catastrophic events, particularly in populated coastal areas such as the MRD [5–7].

Furthermore, the combined effects of rising sea levels (~9.16 mm/year at the present) [8,9], subsidence [10,11], reduced sediment supply [12], and hurricanes are jeopardizing the stability of the MRD [13,14], whereas the shorelines are retreating at an alarming rate of up to 14 m/year [5,6,15,16]. The prevalent longshore currents carry much of these eroded sediments west from the MRD and deposit it along the Gulf coastlines in west Louisiana and east Texas [17]. To combat the coastal erosion and strengthen the Mississippi Delta shoreline, a series of restoration projects was implemented [18–20]. More recently, a beach nourishment project was undertaken by the Louisiana Coastal Protection and Restoration Authority (CPRA) in 2013 to replenish ~$2.8 \times 10^6$ m$^3$ of sediments along ~10 km of shoreline near Port Fourchon (on the west side of the Caminada–Moreau headland) [21]. In addition, sand fences and dune vegetation (*Panicum amarum* and *Uniola paniculata*) were installed and planted along the supratidal zone to control the aeolian erosion and increase sand deposition [6,22,23]. Have these coastal restoration efforts been able to withstand the natural and anthropogenic disturbances and subsequentially protect the coastal wetlands (e.g., saltmarsh and mangrove) over the years, especially against major hurricanes such as Ida and Zeta? This question has not been adequately addressed, yet it has significant scientific implications, since, globally, sandy beach coastlines such as the one examined in this study have been under increasing threats of shoreline retreat and erosion [24,25]. Thus, large data gaps exist in the literature regarding the coastal stability at the MRD.

To fill these data gaps and provide the first look at Zeta and Ida's damage on the MRD, we investigated the coastal zones at Port Fourchon, the ground zero of Hurricane Ida at landfall (Figure 1). This study presents a high-resolution spatial-temporal analysis incorporating planialtimetric data acquired via LIDAR, drone, and satellite to investigate the shoreline changes and sand dune dynamics at Port Fourchon from 2002 to the present. The overarching goal of this study is to provide a robust baseline dataset for future coastal restoration efforts in hurricane-prone subtropical coastal regions. The two specific objectives are to (1) reveal the shoreline dynamics near Port Fourchon before and after the beach nourishment project; (2) investigate the impacts of Hurricane Ida and Zeta on the MRD shoreline.

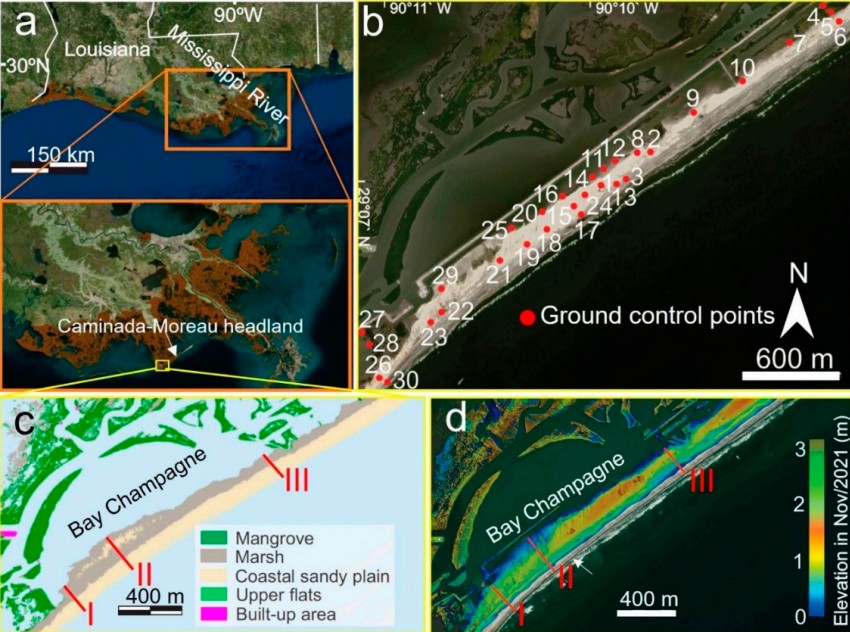

**Figure 1.** (**a**) Satellite image showing the study region; (**b**) Orthoimage marking the locations of the 30 ground control points; (**c**) Vegetation units and (**d**) Digital Elevation Model of the study area (November 2021). The red lines mark the cross-shore profile I–III. The GPS of each ground control point is listed in Table 1.

**Table 1.** Ground Control Points of the study area with longitude, latitude, elevation, and differences (m) between latitudes, longitudes, and elevation obtained by photogrammetry and those obtained in the field by a topographic survey.

| GCP | X/Longitude | Y/Latitude | Z/Elevation (m) | $X_{DIF}$ | $Y_{DIF}$ | $Z_{DIF}$ |
|---|---|---|---|---|---|---|
| 1 | −90.169318 | 29.117012 | 0.78 | −0.014713 | 0.000967 | 0.04 |
| 2 | −90.166061 | 29.119106 | 0.72 | 0.004428 | 0.004298 | 0.01 |
| 3 | −90.167760 | 29.117485 | 0.45 | −0.003994 | 0.000603 | 0.04 |
| 4 | −90.154767 | 29.127904 | 0 | 0.066736 | 0.03 | 0.13 |
| 5 | −90.154317 | 29.127531 | 0.45 | 0.088765 | 0.02 | 0.10 |
| 6 | −90.153269 | 29.126278 | 0 | 0.088842 | 0.003677 | 0.038 |
| 7 | −90.157144 | 29.124695 | 1.13 | 0.065826 | 0.009959 | 0.03 |
| 8 | −90.166814 | 29.119035 | 0.6 | 0.034 | 0.303264 | 0.09 |
| 9 | −90.164600 | 29.120375 | 1.18 | 0.018295 | 0.006667 | 0.03 |
| 10 | −90.161136 | 29.122167 | 1.03 | 0.039914 | 0.006835 | 0.03 |
| 11 | −90.168978 | 29.118352 | 0.27 | −0.008730 | 0.008301 | 0.03 |
| 12 | −90.168500 | 29.118745 | 0.21 | −0.005413 | 0.009216 | 0.03 |
| 13 | −90.168125 | 29.117093 | 0.20 | −0.008472 | −0.000703 | 0.02 |
| 14 | −90.170295 | 29.117766 | 0.3 | 0.014279 | −0.005867 | −0.01 |
| 15 | −90.171358 | 29.115810 | 1.0 | 0.002286 | 0.003633 | −0.02 |
| 16 | −90.172034 | 29.116723 | 0.61 | 0.003032 | 0.004055 | 0.003 |
| 17 | −90.170875 | 29.115429 | 0.26 | −0.000194 | 0.005523 | −0.001 |
| 18 | −90.173615 | 29.114195 | 0.75 | 0.002720 | 0.011527 | −0.04 |
| 19 | −90.176311 | 29.112635 | 0.78 | 0.006339 | 0.016190 | 0.003 |
| 20 | −90.173905 | 29.115531 | 0.87 | −0.022920 | −0.040214 | −0.002 |
| 21 | −90.178845 | 29.110735 | 1.15 | 0.042 | 0.046 | −0.02 |
| 22 | −90.181842 | 29.108411 | 0.78 | −0.636392 | 0.054 | −0.04 |
| 23 | −90.181485 | 29.108824 | 0.54 | 0.19 | 0.053 | 0.02 |
| 24 | −90.169279 | 29.117025 | 0.78 | 0.008734 | −0.000714 | 0.03 |
| 25 | −90.176196 | 29.114711 | 0 | −0.000529 | −0.003827 | 0.05 |
| 26 | −90.181666 | 29.109986 | 0 | 0.001820 | −0.002098 | 0.08 |
| 27 | −90.182765 | 29.111802 | 0 | −0.006467 | −0.008261 | 0.07 |
| 28 | −90.182708 | 29.111594 | 0 | 0.005128 | 0.006270 | 0.03 |
| 29 | −90.180144 | 29.111834 | 0.78 | −0.130523 | −0.783544 | 0.05 |
| 30 | −90.181499 | 29.109870 | 0.58 | 0.769515 | −0.681291 | 0.08 |

## 2. Geographic Background

### 2.1. Study Area

Our study area is located in Bay Champagne (between 29°09′–29°06′N and 90°11′–90°08′W) and the adjacent beach barrier (Figure 1). Bay Champagne is a shallow backbarrier lagoon situated to the southeast of Port Fourchon, ~80 km to the west of the Mississippi bird-foot delta. It is ~2 km² in size with a maximum depth of 2.5 m and an average salinity of ~30‰ [11]. The shoreline near Bay Champagne has retreated by ~2200 m since AD 1887 and has been retreating at ~10 m/year during the recent decades [16,26–28]. Currently, Bay Champagne is separated from the Gulf of Mexico (GOM) by a sand barrier of ~1–2 m above sea level (Figure 1d). The bay is surrounded by saltmarshes (*Spartina alterniflora*) and stunted (mostly < 1 m tall) black mangroves (*Avicennia germinans*), with a saltmarsh-

mangrove ecotone on the north side of the lagoon [6] (Figure 1c). The beach barrier at Bay Champagne was regularly breached by storm surges caused by major hurricanes, including Lili (2002), Katrina (2005), Gustav (2008), Zeta (2020), and Ida (2021) [29,30].

### 2.2. Meteorological Data of Hurricane Ida and Zeta

Since the beginning of the 21st century, four major hurricanes have made landfall in close proximity to our study area [4]. The most recent one, Hurricane Ida, was formed in the southwest Caribbean Sea on 23 August 2021, and it rapidly intensified into a Category 1 hurricane on 27 August while making its way into the GOM across Cuba [3]. On 29 August 2021, the 16th anniversary of Hurricane Katrina, Ida made landfall at Port Fourchon, Louisiana as a Category 4 hurricane (Figure 2). At landfall, Ida had maximum sustained winds of ~240 km/h and caused up to 5 m of storm surges at Port Fourchon [3]. It was the 2nd and 5th strongest landfalling hurricane in Louisiana and the continental U.S. on the record. The severe winds, high storm surges, and extensive floods caused catastrophic damage to the coastal town of Port Fourchon and Grand Isle near the Mississippi Delta (Figure 2b–e). Ida also broke several rainfall records while moving across the USA Northeast. Overall, Ida caused over $65 billion in damages and a total of 115 deaths, including 33 total deaths in Louisiana [3].

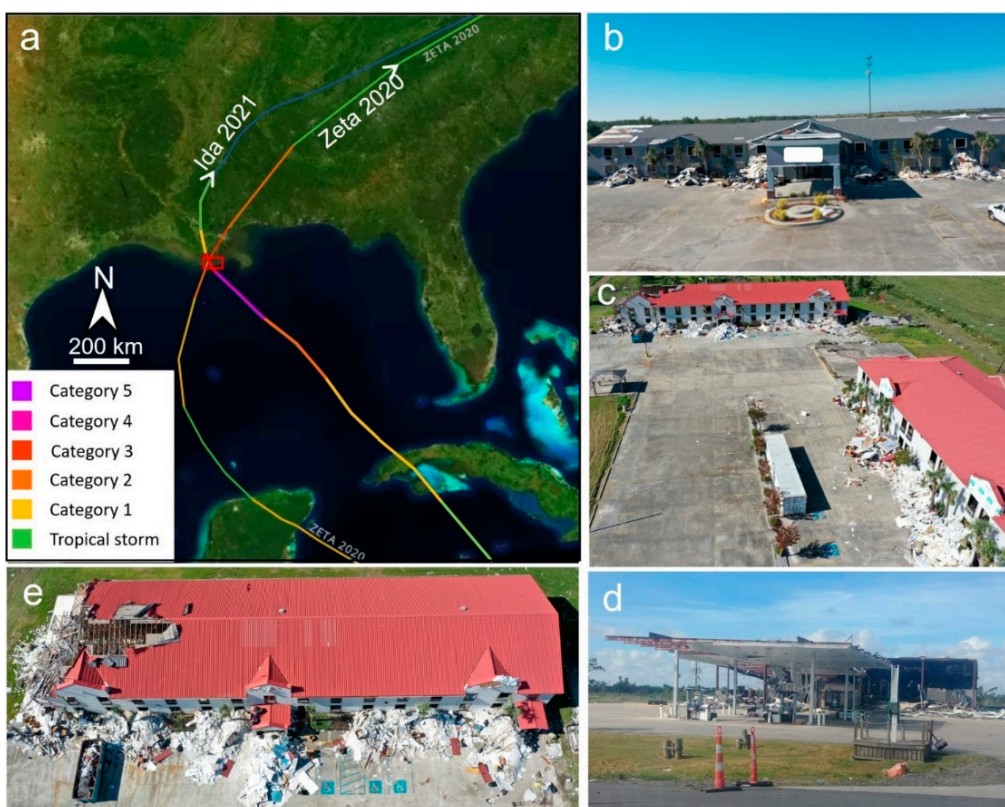

**Figure 2.** (**a**) Storm tracks (https://coast.noaa.gov/hurricanes, accessed on 10 May 2022) of Hurricane Ida and Zeta. The red box marks our study area, the coastal town of Port Fourchon. (**b–e**) Drone images showing Ida's impact on Port Fourchon.

Approximately ten months prior to Ida, another major hurricane, Zeta, made landfall at Cocodrie, Louisiana, ~50 km to the west of our study area, on 28 October 2020 (Figure 2). Zeta was a Category 3 hurricane at landfall, with maximum sustained winds of ~185 km/h. It caused ~3 m of storm surge inundation to the east of the MRD, especially in Mississippi and Alabama, and 1 to 2 m of storm surges at Port Fourchon [31].

## 3. Methods and Materials

### 3.1. Remote Sensing

The remote sensing analysis of the sand barrier dynamics at Bay Champagne utilized planialtimetric data and followed a three-phase methodology flow chart (Figure 3): (1) a spatial-temporal analysis based on satellite images and LIDAR data; (2) the production of a digital elevation model (DEM) based on ground-validated drone photogrammetry; (3) data integration to evaluate the impacts of Hurricane Ida on the beach barrier. All of the drone missions were conducted during low tide. More details regarding the three-phase methodology can be seen in [7].

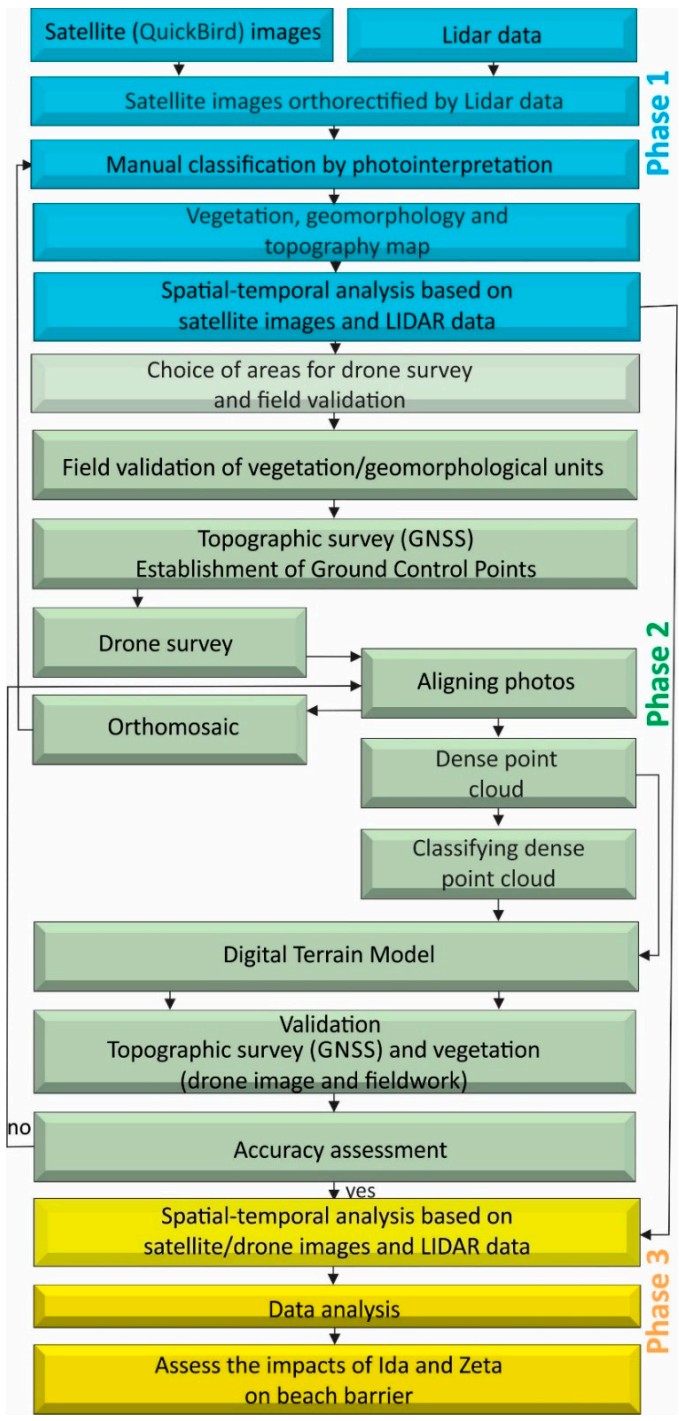

**Figure 3.** Methodological flow-chart of the spatial-temporal analysis.

Every remote sensing image was precisely orthorectified by the LIDAR model. Quick-Bird images (2.44 m multispectral ground pixel resolution with blue, green, and red band) of October/2002, April/2010, March/2013, October/2020, and September/2021 were processed with Agisoft Metashape Professional v.1.6.2. The LIDAR data for years 2002 (NOAA/NASA/USGS), 2010 (JALBTCX), and 2013 (USGS/NASA) were retrieved from the Atlas website from Louisiana State University (https://atlas.ga.lsu.edu/) (accessed on 21 November 2021) and National Oceanic and Atmospheric Administration (NOAA) website (https://coast.noaa.gov/dataviewer/#/) (accessed on 21 November 2021), and they had a horizontal and vertical accuracy of 73–100 cm and 10–15 cm, respectively. LIDAR data provided planialtimetric data to categorize the study area into ground, water, and vegetation.

### 3.2. Spatial-Temporal Analysis

The spatial-temporal analysis of years 2010 (before beach nourishment), 2013 (after beach nourishment), 2019 (before Zeta), 2020 (after Zeta), and 2021 (after Ida) was processed in the Global Mapper v.22.1 (Blue Marble Geographics, Hallowell, ME, USA). The planialtimetric analysis was completed with high-resolution images (2.6 cm/pixel) of November/2019 and November/2021 using Phantom 4 Advanced drone. Ground surveys were conducted during the drone surveys to validate the topography, vegetation heights/types, and shoreline dynamics between November/2019 and November/2021. Topographic surveys on the ground allowed the determination of the supratidal and intertidal zone limits during that time interval. The planialtimetric cross-shore profiles I–III (Figure 1c,d) were retrieved by using an electronic Self-Leveling Horizontal Rotary Laser, a Trimble Catalyst receiver, and a differential GPS system with decimeter correction (horizontal and vertical precision ± 10 cm). These absolute planialtimetric data, based on GEOID18, were used as Ground Control Points (GCPs, a total of 30 points) to improve and evaluate the image positions and to calibrate the DEM (Table 1). Sediment volumes were measured within a designated zone according to the elevation grid generated for each drone and Lidar survey, relative to a baseline, defined as mean sea-level (0 m) using Global Mapper software version 22.1. Volumetric calculations were executed by dividing the area of interest into small rectangular pieces following a uniform grid and then calculating the sum volume of the small 3D rectangles (Volume = Height $*$ Pixel Size) between terrain models and the cut surface [32].

The vertical differences ($Z_{dif}$) between low and high tide were less than 13 cm, revealing a vertical margin of error of $\pm 13$ cm for the 3D models. The planimetric differences (latitude and longitude) were <0.78 m (Table 1). Considering the $X_{dif}$, $Y_{dif}$, and $Z_{dif}$ values, margins of error were estimated at $\pm 0.079$ m$^3$ and $\pm 0.15$ m$^3$ for the volume calculations based on drone and Lidar data (vertical and horizontal accuracy of 15 and 100 cm), respectively.

GPSs of landmarks (e.g., bridges and houses) were used as stable reference lines for spatial-temporal analysis. The contour line of 0 m above the mean sea level was used as the lower limit of the intertidal zone to calculate the shoreline dynamics. Distance measurements were obtained by Global Mapper v.22.1 on georeferenced satellite and drone images. Considering the ground pixel resolution of 2.44 m of the QuickBird images, a margin of error of 2.5 m was estimated for the planimetric data obtained from these images. The vegetation and geomorphological features were manually classified by photointerpretation using the Global Mapper software. The dataset of sites with a previously identified land cover was applied to define the image features (multispectral digital numbers) related to the texture of drone orthoimages (spectral information between 380 and 710 nm) of each land cover type. Drone images (resolution of 2.6 cm) permitted the identification of mangroves, saltmarshes, and the sandy coastal barrier. Drone panoramic aerial photos were also used to identify the vegetation and geomorphological units. More details regarding drone image processing and 3-D model construction are described in the Supplementary Content Text S1.

## 4. Results

### 4.1. Beach Barrier Dynamics between 2019 and 2021

Three cross-shore profiles were established to compare the beach barrier dynamics during the study period (Figure 4a,b). Each profile was marked between two fixed reference points to calculate the vertical and horizontal shoreline change. Substantial changes were observed along all three cross-shore profiles, especially after the landfalls of Hurricanes Zeta and Ida (Figure 4c,d).

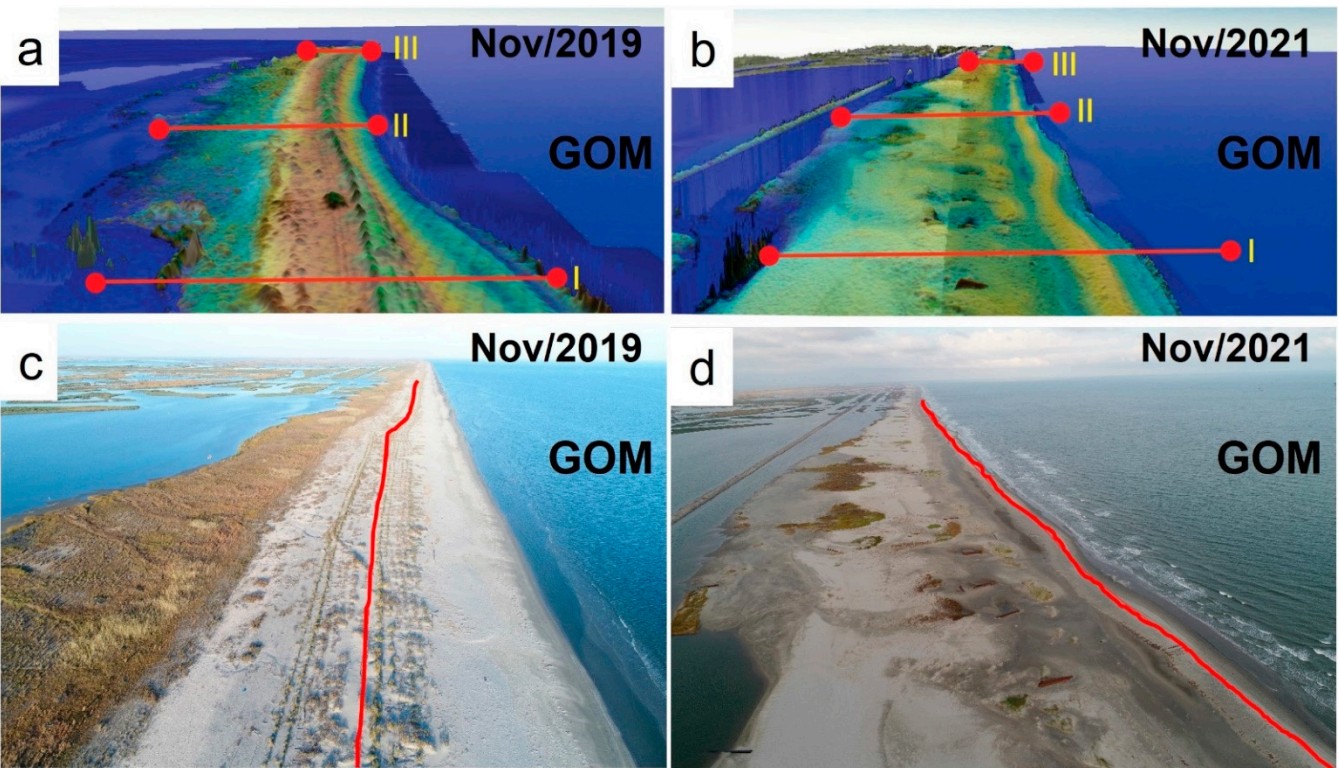

**Figure 4.** DEMs of the beach barrier in (**a**) 2019 and (**b**) 2021 (see location in Figure 1d). Drone images (taken during low-tide) of the studied beach barrier in (**c**) 2019 and (**d**) 2021. The red dots in (**a**,**b**) mark the fixed reference points that establish the three cross-shore profiles. The red lines in (**c**,**d**) mark the dune crest position in 2019.

Planialtimetric data (LIDAR and drone) show significant vertical and horizontal displacements of the dune crests between 2019 and 2021 along all three cross-shore profiles. The maximum dune heights in 2019 along the profiles I, II, and III were ~2.3, 2.2, and 2.1 m, but these numbers were reduced to ~1, 0.9, and 1 m in 2021, respectively (Figure 5). The average dune height was reduced by over 1 m after the two consecutive hurricane landfalls (Figure 5). Meanwhile, substantial erosion occurred along the sandy intertidal flats at the seaward edge of the beach barrier. For instance, the shoreline retreated for 62, 25, and 64 m along the profiles I, II, and III between 2019 and 2021 (Figure 5). Subsequently, the intertidal and supratidal portions of the cross-shore profiles (sum of the red areas in Figure 5) were reduced by 108,434 m$^2$ after Hurricane Zeta (between November/2019 and October/2020) and by 151,030 m$^2$ after Hurricane Ida (between October/2020 and September/2021). At the other end of the beach, the sandy backbarrier flats gained ~34,000 m$^2$ between November/2019 and October/2020 and ~100,872 m$^2$ between October/2020 and September/2021 (sum of the yellow areas in Figure 5).

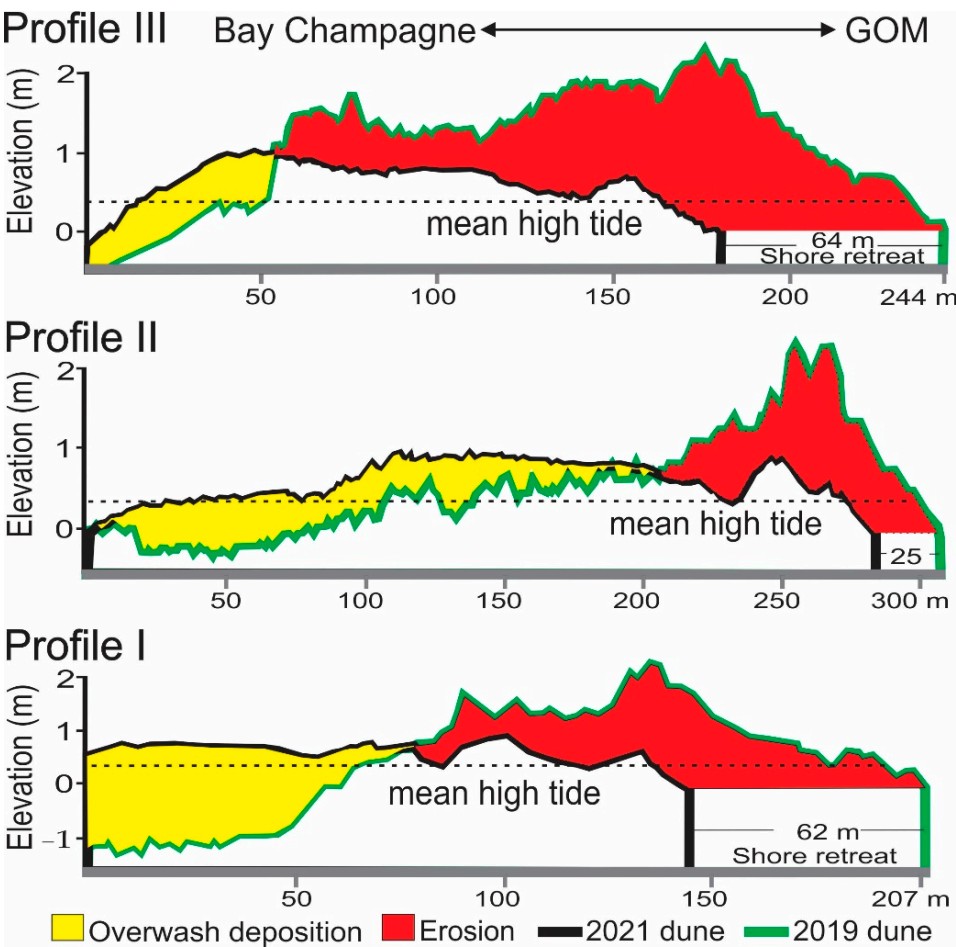

**Figure 5.** Planialtimetric cross-shore profiles I, II, and III (see location in Figure 1d) based on drone data. The dune crest positions along the three profiles were marked by green line in November/2019 and by black line in November/2021. The red and yellow areas mark the erosion and redeposition of the beach barrier sediments.

*4.2. Spatial-Temporal Analysis of the Beach Barrier between 2002 and 2021*

The total sediment volume of the beach barrier in our study area fluctuated between 2002 and 2021 (purple boxes in Figure 6). Before the CPRA beach nourishment project, the total floor area (the surface area of the studied beach barrier in the purple box) and the volume of the studied barrier were 32 ha and 329,628 m$^3$, respectively, in 2002 (Figure 6a). These numbers changed to 50.1 ha and 275,166 m$^3$ in 2010 (Figure 6b). Overall, the total floor area of the studied barrier gained 18.1 ha (+56.6%), but its volume was reduced by 54,462 m$^3$ ($-16.5$%) between 2002 and 2010. In particular, the average dune crest height was reduced by over 1 m (Figure 6a,b). The three red lines in Figure 6 mark the distance between three fixed reference points and the seaward edge of the beach barrier at the low tide. These lines were used to measure and compare the shoreline dynamics along the three cross-shore profiles during the study period. Between 2002 and 2010, the shoreline at cross-shore profiles I, II, and III retreated 142 m ($-17.2$%), 157 ($-13.2$%), and 210 m ($-23.5$%), respectively (Figure 6a,b). On average, the shoreline in the study area was retreating at a rate of 21.2 m/year between 2002 and 2010, over 50% higher than the previous estimation of $12-14$ m/year between 1983 and 2014 (Yao et al., 2018; Cohen et al., 2021a; Dietz et al., 2018). Thus, the shoreline retreat was rapidly accelerating during the past decade.

At the start of the beach nourishment effort, the overall floor area and volume of the studied barrier were 41 ha and 304,583 m$^3$ in 2013 but then increased to 62.3 ha (+52%) and 646,533 m$^3$ (+112.3%) in 2019, respectively (Figure 6c,d). Meanwhile, the shoreline at profiles I, II, and III advanced 42 m (+6.2%), 56 m (+5.6%), and 40 m (+6.1%) between 2013 and 2019,

respectively (Figure 6d). After being directly hit by two major hurricanes (Zeta and Ida) in consecutive years, these numbers decreased to 50.6 ha (−18.8%) and 405,675 m³ (−37.3%) between 2019 and 2021, while the shoreline retreated 62 m (−9.1%), 24 m (−2.4%), and 64 m (−10.9%) at profiles I, II, and III, respectively (Figure 6e). On average, the shoreline was retreating at a rate of 25 m/year between 2019 and 2021, 18.4% higher than that prior to the beach nourishment effort.

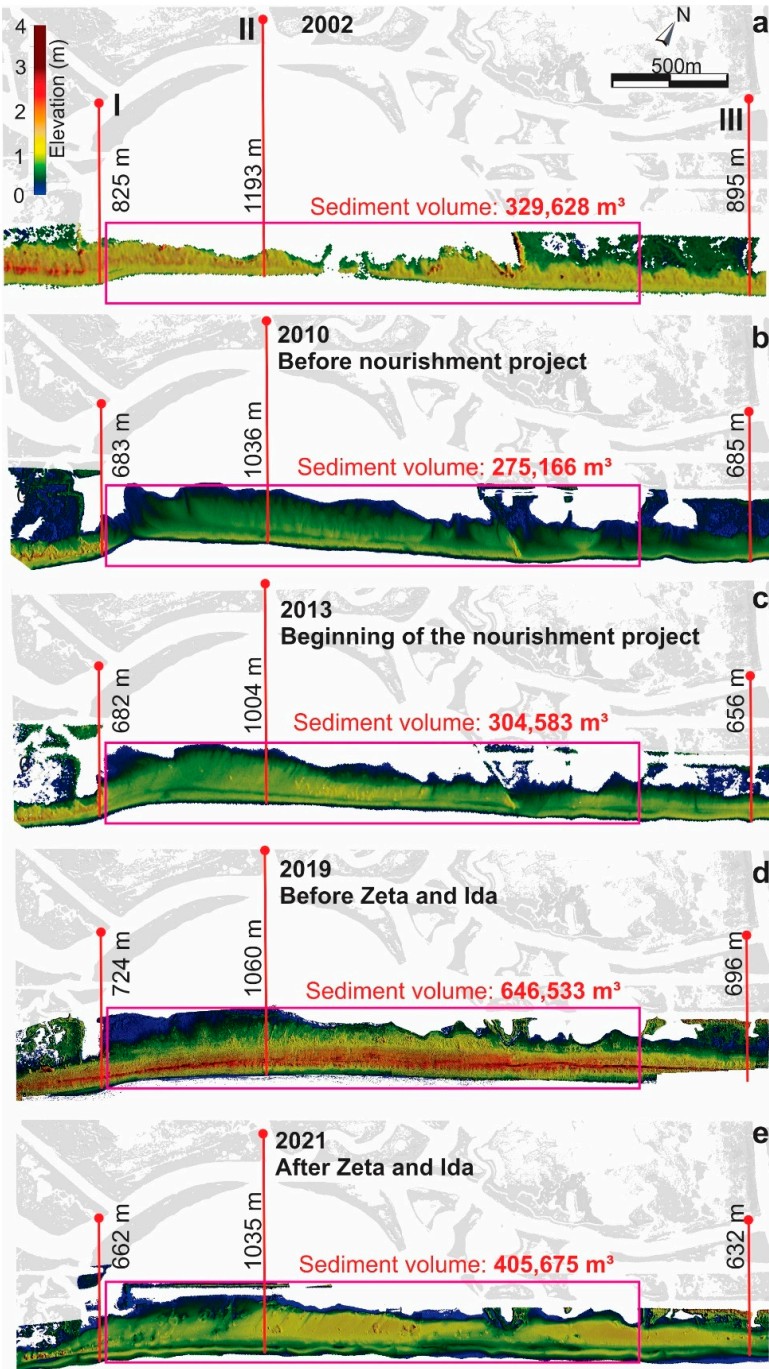

**Figure 6.** Spatial-temporal analysis of the beach barrier dynamics based on satellite and drone images between 2002 and 2021 (**a–e**). The purple boxes mark the areas used as references for sediment volume calculation. The red dots mark the three fixed reference points that were used to calculate the shoreline dynamics at planialtimetric cross-shore profiles I, II, and III. The three red lines in each figure mark the distance between the fixed reference points and the seaward edge of the beach barrier at the low tide.

## 5. Discussion

### 5.1. Shoreline Dynamics at Bay CHAMPAGNE Prior to Hurricanes Ida and Zeta

An analysis of the remote sensing images revealed dramatic shoreline fluctuations near Bay Champagne (Figure 6). Prior to the beach nourishment project, the shoreline in the study area retreated ~170 m (−18%) between 2002 and 2010 (average values of profiles I, II and III). However, after the beach nourishment project in 2013, the shoreline advanced (on average) ~46 m (6%) between 2013 and 2019. Thus, the remote sensing analysis indicates that the restoration effort was effective during the study period.

From a long-term perspective, historic maps from the Louisiana State University Cartographic Information Center show that Bay Champagne was a circular, inland lake with a direct tidal inlet connecting to the Gulf of Mexico in the late 19th century (Figure 7a). At that time, the lake was located approximately one kilometer inland from the GOM. By 1953, the shoreline had retreated to the edge of the lake, while the tidal inlet disappeared (Figure 7b). By 1978, nearly half of the original lake area had disappeared, and the shoreline moved landward for 500 m (Figure 7c). Over the next 32 years, two-thirds of the original lake area had disappeared, and the shoreline retreated another 240 m (Figure 7d). Thus, although the above shoreline movements were estimated based on a single reference point, it is clear that shoreline erosion was a continuous phenomenon along the studied beach barrier since the late 19th century, in line with previous studies of the Caminada–Moreau headland [15,16,28].

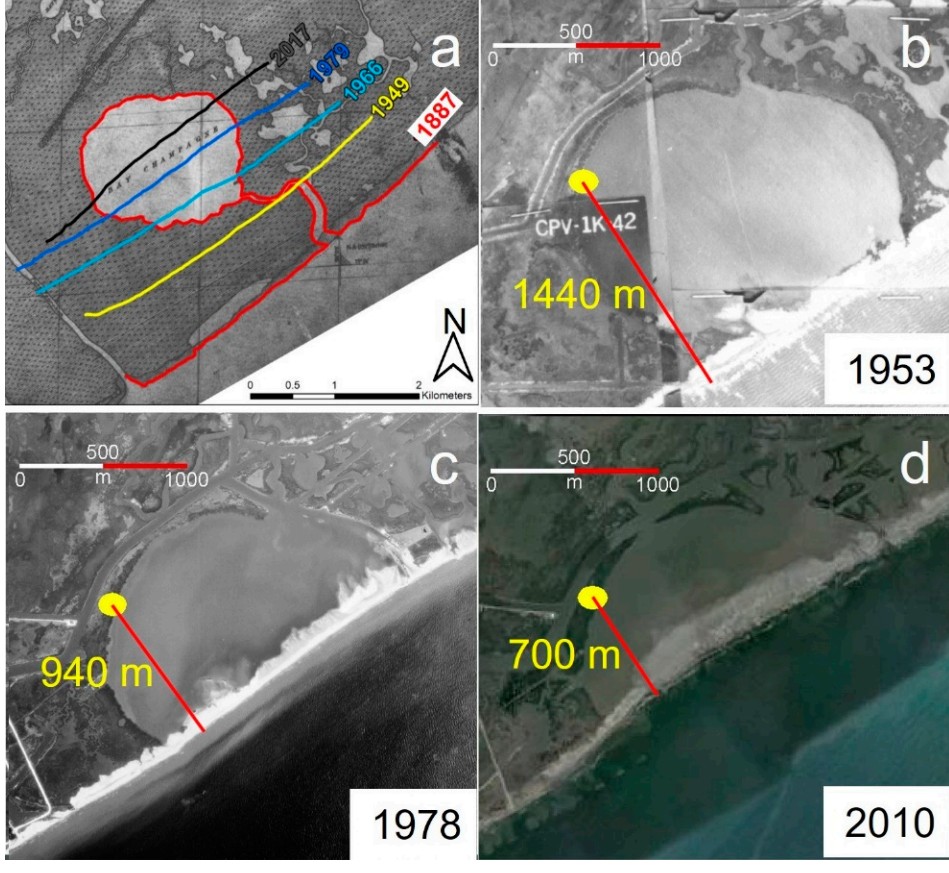

**Figure 7.** (**a**) Historic map (Dietz et al., 2022), (**b**,**c**) aerial photographs, and (**d**) satellite image showing rapid shoreline retreat at Bay Champagne since the late 19th century. The red lines perpendicular to the Shoreline show the distance between a fixed reference point (yellow circle) and the seaward edge of the beach on every satellite image. These lines were used as references to measure the shoreline retreat. Historic map and aerial photographs were retrieved from the Louisiana State University Cartographic Information Center.

Located to the west of the current Mississippi River Delta, the Caminada–Moreau headland was once part of the active Mississippi River Delta that was abandoned ~500 years ago when the delta lobe switched to the east [33–35]. Since then, erosional processes have been persistent and have caused significant land loss at the Caminada–Moreau headland [15,36], making it the region with the highest rate of shoreline retreat around the globe [36–38]. However, as our spatial-temporal analysis indicates (Figure 6c,d), CPRA's beach nourishment project (including sand replenishment, sand fence installation, and dune vegetation planting) had successfully stopped coastal erosion and stabilized the shoreline near Bay Champagne between 2013 and 2019, perhaps for the first time since the delta lobe was abandoned.

### 5.2. Hurricane Effects on Beach Barrier Dynamics

Before CPRA's beach nourishment project, the volume of this ~2 km long beachfront was disappearing at a rate of ~6808 $m^3$/year as a result of human and natural disturbances, and the shoreline was retreating at a rate of ~21 m/year (the average change of the three cross-shore profiles) between 2002 and 2010 (purple box in Figure 6a,b), when two intense hurricanes, Katrina (Category 4, 2005) and Gustav (Category 3, 2008), made landfalls nearby [29,39,40]. After the beach nourishment effort in 2013, the beachfront volume and shoreline width were increasing at a rate of ~56,992 $m^3$/year and ~15 m/year between 2013 and 2019 (Figure 6c,d), despite the frequent impacts of minor tropical storms but no major hurricane landfalls [4].

Over the past two decades, the major physical mechanisms driving coastal erosion at Bay Champagne, such as the relative sea-level rise and deltaic sediment compaction, have remained relatively constant [16,28]. Moreover, Atlantic hurricane activity has fluctuated during the 20th and early 21st centuries [41–44], and periods of frequent landfalling hurricanes are known to accelerate coastal erosion near the MRD [16]. Therefore, Hurricanes Katrina and Gustav have likely contributed to the rapid coastal erosion in our study area during the first decade of the 21st century (Figure 6a,b). However, the sand fences and planted dune vegetation have evidently reduced aeolian erosion and increased sand deposition after 2013. Thus, CPRA's restoration effort in our study area has effectively stabilized the coastal zone during the hurricane-quiescent years between 2013 and 2019.

Furthermore, after the two hurricane strikes by Hurricanes Zeta and Ida in consecutive years (2020 and 2021), the volume of the beachfront was reduced by 240,858 $m^3$, almost 40% of its volume in 2019, while the shoreline retreated ~60 m (Figure 8). Moreover, the average height of the dune crest across the studied beachfront was reduced by over 1 m (Figure 4c,d), while the transport of overwash sand (82,915 $m^3$) to backbarrier environments was recorded after both Zeta and Ida (Figure 8). All the evidence points to the removal and redistribution of beach sediments during these hurricane strikes [6,45]. Thus, it is clear that this beach nourishment effort was not able to withstand the direct landfall of intense hurricanes. If we consider the ~20-year return period for major (Categories 3−5) hurricane strikes in this area (NOAA, 2021b) and 40% loss of the beach volume after 2 major hurricane strikes, it is reasonable to assume that the entire beach barrier will be removed after several more major hurricane strikes. Moreover, Bay Champagne has lost over 45% of its surface since the 1970s as a result of long-term shoreline retreat [16]. Thus, our estimates support the conclusion that, at the current rate of shoreline recession averaged over the past four decades, Bay Champagne will disappear completely into the Gulf of Mexico within the next 40 years, if no further human intervention is undertaken [16,28]. Hence, it is possible to conclude that on a decadal timescale, the current restoration effort may not be adequate to compensate for the coastal erosion in this part of the MRD due to the likelihood of more frequent major hurricane strikes in the future [1].

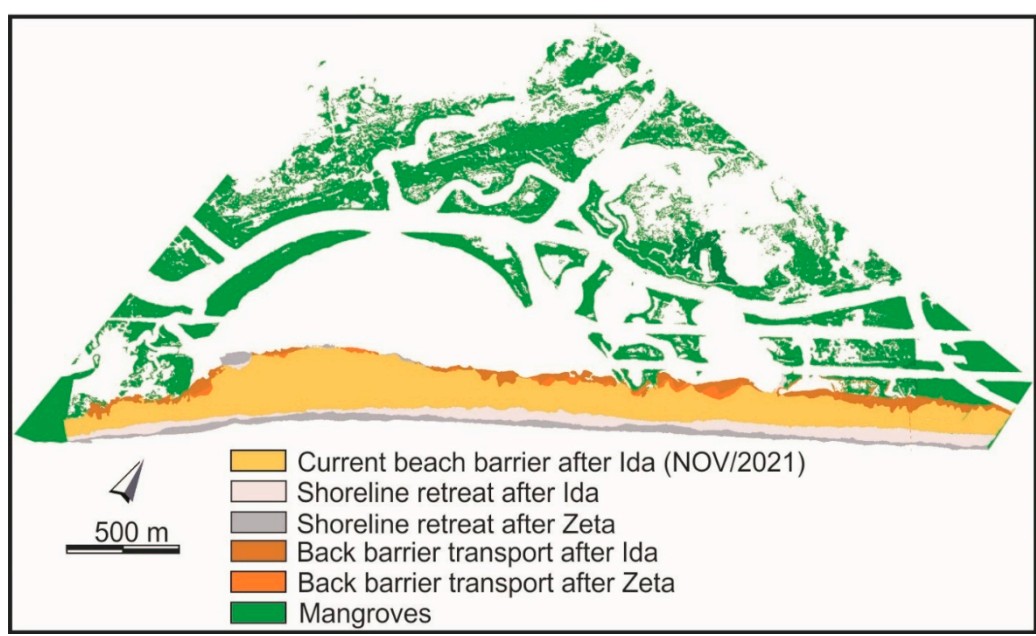

**Figure 8.** Composite diagram showing beach barrier dynamics associated with Hurricane Ida and Zeta.

*5.3. Implications for Future Coastal Restoration Efforts*

At the MRD, anthropogenic activities have become a primary factor of the land loss problem in the region. The construction of >50,000 dams and levees upstream of the Mississippi River not only has effectively stopped the natural overbank flooding, but also prevented river sediments from replenishing the wetlands [13,46]. Building of waterways and canals to support the petroleum industries has also led to increased saltwater intrusion and wetland fragmentation [39,47,48]. Moreover, the impoundments and other engineering activities such as soil erosion controls, bank revetments, and meander cutoffs have led to ~70% reduction of the fluvial sediment load since the 1850s [12,49–52] As a result of these human activities, the MRD has lost ~5000 km$^2$ of coastal wetlands and ~75% of the land area since the 1930s [20].

Figure 9 compares the beach barrier before and after Hurricanes Ida and Zeta. It is clear that the sand fences and dune vegetation were not able to withstand the two major hurricane landfalls (Figure 9). As the beach nourishment efforts were demolished by storms, Port Fourchon would once again become increasingly vulnerable to future hurricane impacts. In particular, when the beach nourishment efforts were intact, only direct landfalls of major hurricanes would have breached the beach barriers and transported marine water and sediment into Bay Champagne, as we discussed in the previous sections. After the average dune crest height was reduced by over 1 m (Figures 5 and 7), Bay Champagne would have become more susceptible to marine inundation through storm surge overwash processes [53]. As this cycle continues, our study area is expected to become more suscep­tible to hurricanes and human impacts, further accelerating the rate of shoreline retreat. What is the alternative solution for the rapidly retreating coast in the MRD? The short-term answer will be periodically repeating the beach nourishment efforts, especially after intense hurricane strikes. However, as the root of this coastal land loss issue is anthropogenic activities in the face of climate change and the rapid sea-level rise (~9.16 mm/year) in the area [9], repetitive beach nourishment efforts would become increasingly expensive and futile. Thus, Louisiana's coastal restoration effort is facing a dilemma that may ultimately lead to a tipping point.

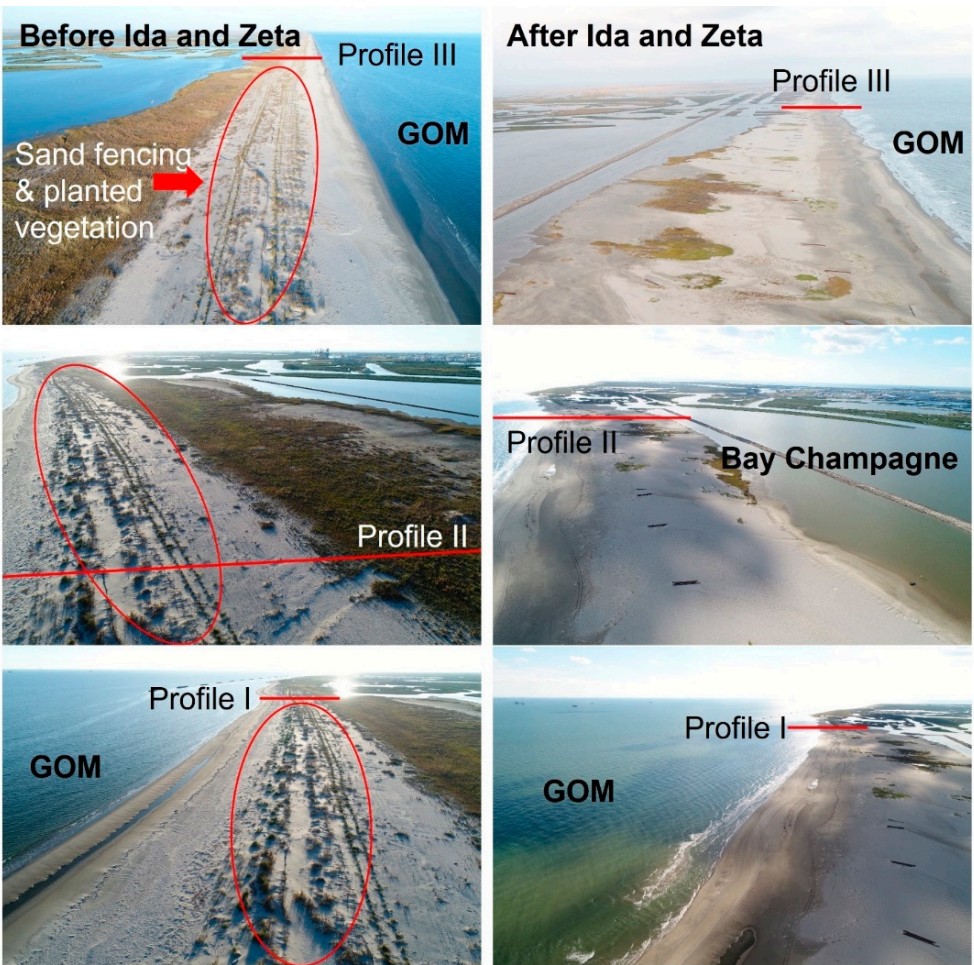

**Figure 9.** Georeferenced drone images comparing the beach barriers near the three cross-shore profiles before and after Hurricane Ida and Zeta. The red circles mark the sand fences and planted dune vegetation prior to the two hurricanes.

An ongoing natural process that may help to reduce the rate of coastal erosion is the proliferation of mangroves in this part of the MRD. As demonstrated by previous studies, mangroves in the MRD are very resilient in withstanding intense hurricane impacts and in recovering from winter freeze damages [6,7,54,55]. More importantly, research from across the world has demonstrated that mangroves can stabilize the coastal zones by attenuating the wind and waves from storms and increasing vertical accretion in the coastal zone by trapping more sediments [56–58]. In fact, significant mangrove expansion has been reported near the MRD since the early 21st century due to warmer winters [7,59,60], and mangrove planting has already taken place near the MRD as a means of shoreline protection [61]. Thus, in the long term, such eco-engineering might be a viable solution to help save the Louisiana coast or other rapidly recessional shorelines across the tropics and subtropics.

## 6. Conclusions

This study indicates that the Louisiana CPRA's beach nourishment project (including sand replenishment, sand fence installation, and dune vegetation planting) has successfully stopped coastal erosion and stabilized the shoreline near Bay Champagne during the hurricane-quiescent years between 2013 and 2019. However, this beach nourishment effort was not able to withstand the landfalls of Hurricanes Zeta and Ida in two consecutive years. Thus, over the long term, the current restoration effort may not be adequate to compensate for the costal erosion in this part of the MRD due to the likelihood of more frequent major

hurricane strikes in the future. However, we do acknowledge that this study was conducted immediately after the landfall of Hurricane Ida, whereas it is possible—albeit unlikely—that the studied beach barrier will be stabilized again with the recovery of the planted dune vegetation. Thus, future studies are needed to keep monitoring this dynamic shoreline near the Mississippi River Delta, particularly focusing on the post-hurricane recovery of the beach barrier. Our three-phase methodology (Figure 3) could provide a viable method for post-disaster assessment and an efficient approach to monitor the effectiveness of coastal restoration projects around the globe.

**Supplementary Materials:** The following supporting information can be downloaded at: https://www.mdpi.com/article/10.3390/rs14112598/s1. Text S1: 3-D Models.

**Author Contributions:** Q.Y. designed the study, led the sample collection, and wrote the paper. M.C.L.C. led the remote sensing analysis, contributed to writing and editing, and assisted in the fieldwork and vegetation survey. K.-b.L. directed the project and fieldwork and contributed to data interpretation, writing, and editing. A.V.d.S. assisted in the remote sensing analysis. E.R. assisted in field sampling and laboratory analysis. All authors have read and agreed to the published version of the manuscript.

**Funding:** This research was funded by the U.S. National Science Foundation (grant # 1759715), the Brazilian National Council for Technology and Science (CNPq 07497/2018−6, 403239/2021−4), Research Funding Agency of the State of São Paulo (FAPESP 2020/13715−1).

**Data Availability Statement:** Datasets produced in this article will be stored at the Neotoma Paleoecology Database (https://www.neotomadb.org (accessed on 26 May 2022)) and accessible to the public for free, upon the publication of this study.

**Acknowledgments:** We thank the Louisiana State University Cartographic Information Center for providing the historic map and aerial photographs of the study area. We also thank LSU Libraries for their Open Access Author fund.

**Conflicts of Interest:** The authors declare no conflict of interest.

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
