# Peer review of "Nature versus Humans in Coastal Environmental Change: Assessing the Impacts of Hurricanes Zeta and Ida in the Context of Beach Nourishment Projects in the Mississippi River Delta"

_remotesensing, doi:10.3390/rs14112598_

Round 1

Reviewer 1 Report

The manuscript reports spatial-temporal analysis data on shoreline dynamics using LIDAR, drone and satellite images. I believe the methods used in the study are adequate for this journal, and the results are interesting to the readers in this field.

However, there are several major questions to be answered before accepting the paper for publication.

  1. The flow chart in Figure 3 is too complicated for clear understanding. Please provide more detailed description on each step.
  2. The sediment volumes in Figure 6 were calculated at low tide or high tide ?

I guess the profiles in Figure 5 were calculated at low tide as the drone data were measured at low tides. However, no information can be found when the satellite and LIDAR data were measured (at low tide or high tide) ?   

It is necessary that error ranges on the volumes calculated in Figure 6 are to be estimated considering the tidal range in the study area, if the corresponding information were not available.

  1. Lines 24 – 25 (Abstract), it is said that “Bay Champagne will likely disappear completely into the Gulf of Mexico within the next 40 years”. Please provide the reasons that the authors made this conclusion in more detail.

Minor suggestion

- Mark the location of the study site on Fig. 2(a)

Author Response

Reviewer #1

The manuscript reports spatial-temporal analysis data on shoreline dynamics using LIDAR, drone and satellite images. I believe the methods used in the study are adequate for this journal, and the results are interesting to the readers in this field.

However, there are several major questions to be answered before accepting the paper for publication.

1. The flow chart in Figure 3 is too complicated for clear understanding. Please provide more detailed description on each step.

Authors: We agree with this suggestion that the original Figure 3 shows too many series and parallel connections and is difficult to follow. In the revision, Figure 3 is replaced by a new figure that shows a more linear and coherent step-by-step methodological flow chart. We understand there are still a lot of steps in this new figure, but this study utilizes a sophisticated high-resolution spatial-temporal analysis and these are the essential steps in order to replicate this study. Thus, we believe it is necessary to display all the steps for the readers.

Moreover, we have elaborated on the description of satellite image processing and added more details in lines 162-170 to improve the understanding of the flow chart in Figure 3. In addition, more details of the methodology can be found in the supplementary content.

2. The sediment volumes in Figure 6 were calculated at low tide or high tide?

I guess the profiles in Figure 5 were calculated at low tide as the drone data were measured at low tides. However, no information can be found when the satellite and LIDAR data were measured (at low tide or high tide)?

Authors: All of the drone missions were conducted during low tide. This statement is added in the Methods and Materials (lines 135-136) in the revision. However, the sediment volumes were measured within a designated zone according to the elevation grid generated for each drone and Lidar survey, relative to a baseline, defined as mean sea-level (0 m) using Global Mapper software version 22.1. The absolute elevations are based on GEOID18, obtained by the Trimble Catalyst GNSS receiver used to record the Ground Control Points (GCPs, a total of 30 points). The GCPs were used to improve and evaluate the image positions and to calibrate the DEM  (lines 162 – 170).

3. It is necessary that error ranges on the volumes calculated in Figure 6 are to be estimated considering the tidal range in the study area, if the corresponding information were not available.

Authors: The vertical differences (Zdif) were less than 13 cm, revealing a vertical margin of error of ± 13 cm for the 3D models. The planimetric differences (latitude and longitude) were <0.78 m (Table 1). Considering the Xdif, Ydif, Zdif values, margins of error were estimated at ±0.079 m3 and ±0.15 m3 for the volume calculations based on drone and Lidar data (vertical and horizontal accuracy of 15 and 100 cm for the Lidar data), respectively. These contexts are elaborated in lines 171 – 176 in the revised manuscript.

4. Lines 24 – 25 (Abstract), it is said that “Bay Champagne will likely disappear completely into the Gulf of Mexico within the next 40 years”. Please provide the reasons that the authors made this conclusion in more detail.

Authors: This statement is elaborated in section 5.2 and summarized in lines 329-334.

Minor suggestion

- Mark the location of the study site on Fig. 2(a)

Authors: The location of the study site is marked in that red box on Fig. 2(a) and described in the Figure caption (line 127-128).

Reviewer 2 Report

Review of “Hurricane Ida and Zeta caused significant erosion to the Mississippi Delta shoreline” by Yao et al. submitted to Remote Sensing (remotesensing-1726072).

The manuscript deals with the impact of hurricanes Zeta and Ida on a 2km-long stretch of beach along Port Fourchon, Louisiana. The authors used LIDAR, drone and satellite data to assess beach dynamics since 2002, as well as the impact before and after beach nourishment and the two hurricanes. Results show that the area lost ~40% of sand volume prior to the hurricanes, dune height was reduced and the shoreline retreated by around 60 m. The authors also found that beach nourishment was not sufficient to protect the shorelines of the Mississippi River Delta from intense hurricane landfall. They propose a three-phase methodological approach that can be adapted to coastal monitoring in other places, thus pointed to the significance of this study on a global scale. The manuscript deals with an important topic in disaster mitigation in an area that is not only hit by intense hurricanes, but is also highly affected by sea level rise. Thus, I feel the topic is of great importance and suitable for Remote Sensing, as well as the special issue on "Remote Sensing for Marine Environmental Disaster Response".

I first would like to commend the authors – the manuscript is well written and it has been a while since I have reviewed an article where I have had very little to say. Just a few points, along with those in the annotated manuscript:

English – the entire manuscript should be checked for minor English mistakes. I have addressed some of the issues in the beginning of the manuscript (up to line 67) to illustrate the point.

Title: while the title is an accurate representation of the manuscript, I suggest changing it to something a little more appealing. I think it would attract more attention, but I leave this to the discretion of the authors and editor. Also, since Zeta occurred before Ida, I would switch the order in the title to reflect this.

The Introduction should include a description of meteorological conditions in the area – what are the prevalent wave directions, currents (longshore?), etc.

Figures need some improvement. Fonts are often too big, black scale bars are invisible on dark backgrounds, missing north arrow etc.

I do not see the difference between Fig. 1d and Fig. 4a. Since Fig. 4a is the same as 1d, I would get rid of it and place the Dem above the relevant photo (b above e, c above d).

How were sediment volumes calculated? Based on the differences DEMs (and if so – given the accuracies, can you estimate the error in volume)? Please add to methodology (and not just in the supplementary data)

In the discussion, a paragraph dedicated to the reasons for shoreline retreat and sand loss in the area is needed and not something that is stuck in a paragraph discussing solutions.

Author Response

Reviewer #2

The manuscript deals with the impact of hurricanes Zeta and Ida on a 2km-long stretch of beach along Port Fourchon, Louisiana. The authors used LIDAR, drone and satellite data to assess beach dynamics since 2002, as well as the impact before and after beach nourishment and the two hurricanes. Results show that the area lost ~40% of sand volume prior to the hurricanes, dune height was reduced and the shoreline retreated by around 60 m. The authors also found that beach nourishment was not sufficient to protect the shorelines of the Mississippi River Delta from intense hurricane landfall. They propose a three-phase methodological approach that can be adapted to coastal monitoring in other places, thus pointed to the significance of this study on a global scale. The manuscript deals with an important topic in disaster mitigation in an area that is not only hit by intense hurricanes, but is also highly affected by sea level rise. Thus, I feel the topic is of great importance and suitable for Remote Sensing, as well as the special issue on "Remote Sensing for Marine Environmental Disaster Response".

I first would like to commend the authors – the manuscript is well written and it has been a while since I have reviewed an article where I have had very little to say. Just a few points, along with those in the annotated manuscript:

Authors: We appreciate the positive review #2 gave to this manuscript.

1. English – the entire manuscript should be checked for minor English mistakes. I have addressed some of the issues in the beginning of the manuscript (up to line 67) to illustrate the point.

Authors: The entire manuscript has been proofread again to check for errors and we have also incorporated the edits addressed in the annotated manuscript.

2. Title: while the title is an accurate representation of the manuscript, I suggest changing it to something a little more appealing. I think it would attract more attention, but I leave this to the discretion of the authors and editor. Also, since Zeta occurred before Ida, I would switch the order in the title to reflect this.

Authors: The title has been changed to “Nature versus humans in coastal environmental change: Assessing the impacts of Hurricanes Zeta and Ida in the context of beach nourishment projects in the Mississippi River Delta”.

3. The Introduction should include a description of meteorological conditions in the area – what are the prevalent wave directions, currents (longshore?), etc.

Authors: The description has been added in the revised manuscript per suggestion (lines 57-59).

4. Figures need some improvement. Fonts are often too big, black scale bars are invisible on dark backgrounds, missing north arrow etc.

Authors: Figures 1-6 & 7 have been modified per suggestions.

5. I do not see the difference between Fig. 1d and Fig. 4a. Since Fig. 4a is the same as 1d, I would get rid of it and place the Dem above the relevant photo (b above e, c above d).

Authors: Figure 4 is modified per suggestion.

6. How were sediment volumes calculated? Based on the differences DEMs (and if so – given the accuracies, can you estimate the error in volume)? Please add to methodology (and not just in the supplementary data)

Authors: Sediment volumes were measured within a designated zone according to the elevation grid generated for each drone and Lidar survey, relative to a baseline, defined as mean sea-level (0 m) using Global Mapper software version 22.1. Volumetric calculations were executed by dividing the area of interest into small rectangular pieces following a uniform grid and then calculating the sum volume of the small 3D rectangles (Volume = Height * Pixel Size) between terrain models and the cut surface (Global Mapper User’s Manual, 2020).

The vertical differences (Zdif) were less than 13 cm, revealing a vertical margin of error of ± 13 cm for the 3D models. The planimetric differences (latitude and longitude) were <0.78 m (Table 1). Considering the Xdif, Ydif, Zdif values, margins of error were estimated at ±0.079 m3 and ±0.15 m3 for the volume calculations based on drone and Lidar data (vertical and horizontal accuracy of 15 and 100 cm for the Lidar data), respectively. These contexts are elaborated in lines 162 - 176 in the revised manuscript.

7. In the discussion, a paragraph dedicated to the reasons for shoreline retreat and sand loss in the area is needed and not something that is stuck in a paragraph discussing solutions.

Authors: This paragraph has been added to the revised manuscript as suggested (lines 341-352).

8. I am wondering if there is a downside to this? Do mangroves lift the wind from the beach and change patterns thus affecting other processes? Do the affect longshore transport (is there longshore transport in the area?)

Authors: Mangroves typically grow on tropical coastlines because they cannot survive winter freezes. Because the northern Gulf of Mexico is the latitudinal range limit for mangroves in North America, they do not grow very tall. Historically, the average height for the mangrove population near the Mississippi Delta is ~1.5 m tall. Thus, we don’t believe they will alter any physical processes, and we are not aware of any reports that describe mangrove interference in the longshore transport.

Round 2

Reviewer 1 Report

The manuscript has been revised according to my suggestions, and the the questions were answered adequately.